# Implicitly perturbed Hamiltonian as a class of versatile and general-purpose molecular representations for machine learning

Amin Alibakhshi [1✉] & Bernd Hartke [1]

Unraveling challenging problems by machine learning has recently become a hot topic in many scientific disciplines. For developing rigorous machine-learning models to study problems of interest in molecular sciences, translating molecular structures to quantitative representations as suitable machine-learning inputs play a central role. Many different molecular representations and the state-of-the-art ones, although efficient in studying numerous molecular features, still are suboptimal in many challenging cases, as discussed in the context of the present research. The main aim of the present study is to introduce the Implicitly Perturbed Hamiltonian (ImPerHam) as a class of versatile representations for more efficient machine learning of challenging problems in molecular sciences. ImPerHam representations are defined as energy attributes of the molecular Hamiltonian, implicitly perturbed by a number of hypothetic or real arbitrary solvents based on continuum solvation models. We demonstrate the outstanding performance of machine-learning models based on ImPerHam representations for three diverse and challenging cases of predicting inhibition of the CYP450 enzyme, high precision, and transferrable evaluation of non-covalent interaction energy of molecular systems, and accurately reproducing solvation free energies for large benchmark sets.

[1] Theoretical Chemistry, Institute for Physical Chemistry, Christian-Albrechts-University, Olshausenstr. 40, Kiel, Germany. ✉email: alibakhshi@pctc.uni-kiel.de

Employing machine learning for studying complicated scientific challenges has recently become a widely accepted approach in science. As a continuously growing field, machine learning has become a promising tool in studying many diverse areas in molecular sciences, ranging from major topics in life science such as synthetic biology[1], genomics[2], drug discovery[3], and cell biology[4], to major sub-fields of chemistry such as theoretical[5], organic[6], quantum[7,8], polymer[9], and synthetic[10] chemistry.

Despite this diversity in applications, at the heart of all machine-learning approaches in molecular sciences is a reliance on translating molecular structures to quantities understandable by the machine-learning process. These quantities, which are commonly known as molecular representations in computational chemistry and molecular descriptors in cheminformatics, are uniquely defined and are considered as molecular fingerprints. With the molecular representations defined, the main role of machine learning is then to learn the relationship between those representations and the properties of interest.

The earliest example of employing molecular representations for estimating materials properties, to our knowledge, is the group contribution method proposed in the middle of the last century[11]. This method considers a linear or non-linear dependency between molecular properties and the functional groups present in molecules. The success of this simply defined representation in predicting many properties of chemicals for several decades[12–19] has motivated its employment for approximating potential energies of molecular ensembles[20–23], as one of the most extensively studied and, at the same time, most challenging applications of machine learning in theoretical and quantum chemistry[24,25]. Nevertheless, achieving high accuracy in challenging problems like prediction of interaction energy of molecular systems typically requires more versatile molecular representations. To that end, a number of efficient representations have been proposed and widely used in recent years, such as atom-centered symmetry functions[26], the bispectrum of the neighbor density[27], smooth overlap of atomic positions[28], and the Coulomb matrix[29]. The popularity of these representations mainly stems from their remarkably fast and straightforward acquisition, typically requiring only elementary calculations on the geometrical data of the molecules. An excellent review of the recent progress in employing machine learning for evaluating potential energy has been reported by Manzhos and Carrington[30].

These more advanced representations, despite being more efficient compared to elementary representations like functional group numbers and types, still suffer from the limited utility in studying many challenging and complicated problems of interest.

For machine-learning evaluation of conformational energies via those representations, despite several decades of progress, achieving quantum-chemical accuracy still has typically remained limited to simple molecular systems consisting of very few atom types[26,31–35]. As the other limitation, the application of machine-learning models based on employing the commonly applied representations is usually restricted only to the specific systems for which they were developed. This makes it necessary to re-train those machine-learning models for new systems, which is a highly demanding task. Finally, the currently defined representations mainly follow "non-physical" functional forms which result in very limited extrapolation capabilities[36].

For machine-learning evaluation of solvation free energy via conventional representations, except reporting accurate and promising results in very few recent studies[37,38], for the majority of the other works, the reported accuracies are beyond chemical accuracy, as recently reviewed by us[39]. It demonstrates the challenges of employing conventional representations for machine-learned evaluation of the solvation-free energy.

All these issues imply the necessity and importance of developing novel and more efficient representations, capable of addressing the above-mentioned shortcomings. For that purpose, appropriate representations have to satisfy a number of prerequisites such as invariance with respect to translation or rotation of the origin of the coordinate system, invariance to permutation of atoms of the same element, and yielding a unique or constant number of quantities independent of number and type of atoms in the system[35]. Ideally, representations should also allow transferability of machine-learning models to new systems, generate a large number of quantities to enhance machine-learning efficiency for diverse complicated properties of interest, and be physically interpretable.

With all these considerations, the main aim of the present study is to introduce the Implicitly Perturbed Hamiltonian (ImPerHam) as a new class of molecular representations to fulfill all the above-mentioned requirements.

The general idea behind ImPerHam originates from our recently developed machine-learning model for the evaluation of solvation free energy[39]. In that work, we employed machine learning for a more rigorous integration of the continuum solvation energy components. To that end, in addition to cavity geometrical data, we also employed energy attributes of Hamiltonian perturbed by the implicitly defined solvent as inputs of machine-learning models.

The outstanding efficiency of implicitly perturbed Hamiltonian energy attributes in characterizing the highly challenging case of solvation free energy reported in our recent study[39] motivated us to employ similar quantities computed not only for a single solvent of interest but also for a diverse set of other implicitly defined solvents and use them as molecular representations for other purposes as well.

To clarify the general idea behind the ImPerHam representations, we discuss the evaluation of solvation free energy of methane in water as an example. For that purpose, the conventional approaches require computing energy attributes of the methane Hamiltonian implicitly perturbed in water and integrating them based on the conventional approaches in continuum solvation models or via our recently proposed machine-learning model[39]. Based on the new approach, however, we still employ those energy attributes but computed for methane dissolved a number of other arbitrary real or hypothetic solvents and use them as molecular representations for methane.

An obvious advantage of ImPerHam representations stems from the provided possibility of generating an unlimited number of representations via an unlimited choice of solvents. More importantly, the generated representations are actually thermodynamic quantities attributed to energy and free energy. Considering that based on the foundations of thermodynamics, the majority of system properties and thermodynamic quantities can be expressed as functions of energy and free energy, the representations based on these energy functions can be theoretically related to most properties of interest. For this reason, ImPerHam representations can be expected to perform better than the traditional representations mentioned above, which have less direct and less clear-cut relations to the properties of interest. As another direct result, although an unlimited number of solvents can be defined and employed to generate ImPerHam representations, as will be shown later in the present study, even very few arbitrary but diversely defined solvents can yield efficient machine-learning models for the diverse challenging problems considered in the present study.

The ImPerHam representations can be generated and studied using low-cost quantum-mechanical computations or classical polarizable force fields, which allows fast evaluation of those representations. Nevertheless, the computational cost of generating

the ImPerHam representations can be somewhat higher than for other conventional representations discussed earlier. Despite this higher computational cost of computing ImPerHam representations, considering that the developed models based on them are highly transferable and do not require demanding re-training for new systems, we can still consider them an economic and reasonable choice.

In the present study, we demonstrate the efficiency of the machine-learning models based on ImPerHam representations in studying three challenging and extensively required problems in molecular sciences.

The first case study is the prediction of inhibition of a cytochrome P450 (CYP450) enzyme by small molecules. Prediction of CYP450 inhibition by potential drug candidates is a mandatory consideration in drug discovery, due to the crucially important role of these enzymes in metabolizing the majority of the drugs currently found in the market[40]. Inhibition of CYP450 by potential drugs can result in their accumulation in the body and increase the risk of drug–drug interactions[41]. Mibefradil and Cerivastatin are two examples of commercial drugs which have previously retracted from the market for this reason[42,43]. On the other hand, targeted inhibition of specific cytochromes has been proposed as an effective treatment strategy, as has been demonstrated for metastatic prostate tumors[44]. In the present study, we develop machine-learning models to predict inhibition of CYP450 1A2 as one of the important CYP450 members present in the liver. The structure of human microsomal CYP450 1A2 obtained by Sansen et al.[45] is depicted in Fig. 1.

In addition to inhibition of a CYP450 enzyme by machine-learning, we also investigate the performance of ImPerHam representations in the machine-learning evaluation of solvation free energy as well as non-covalent interaction energy, benchmarked for large datasets. These case studies are not only among the most challenging problems in computational chemistry but also are extensively required in a wide range of scientific fields which makes them ideal for benchmarking the new representations.

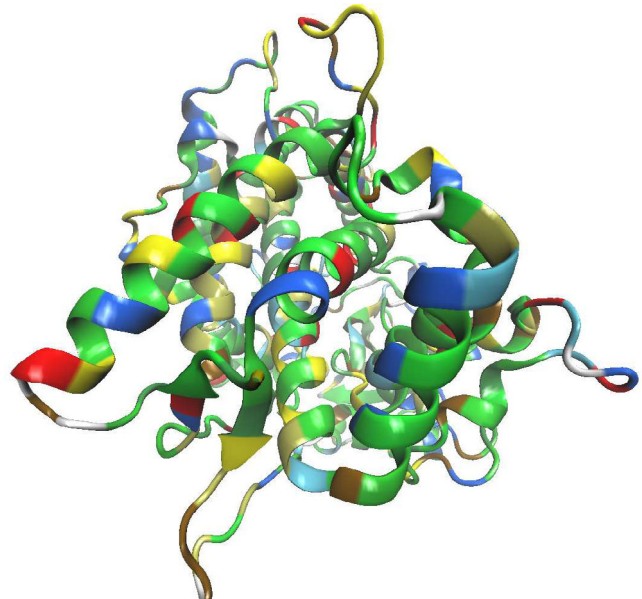

**Fig. 1 Structure of human microsomal CYP450 1A2 enzyme.** Evaluating the possibility of inhibiting this enzyme by drug candidates is one of the early steps in drug design.

## Results and discussion

**Evaluation of CYP450 inhibition**. By training SVM models via 3745 training samples provided by Novotarskyi et al.[46], the predictability of inhibitor activity of 3740 test set molecules was evaluated using the developed machine-learning models. According to the results, the best prediction of inhibitor activity for the training samples was observed for an SVM model that employed 16 representations as model inputs. Via this model, the inhibitor activity of test set compounds could be predicted with 79.6% accuracy which is very close to the MUE of 81.8% reported for the same datasets and machine learning algorithm by Novotarskyi et al.[46]. However, the models studied by Novotarskyi et al. employed a much greater number of representations, ranging from several hundred to roughly five thousand representations. The higher dimension of the representations which simultaneously also results in more parameters for the studied model, typically improves the flexibility of a model to learn complicated mappings, as we demonstrated in a previous study[39]. Nevertheless, at the same time it can also reduce the extrapolation capability of a model[47].

Considering that our employed representations are generated only for a few solvents, it is expected that for further extensions of the number of solvents, a more accurate evaluation of CYP450 inhibition can be achievable.

**Machine-learning approximation of non-covalent interaction energies via ImPerHam representations**. By employing the GFN2-xTB semiempirical method to approximate relative energies of different conformers, we initially obtained an MUE of 22.164 kcal/mol and RMSE of 196.91 kcal/mol compared to the CCSD(T) reference energies.

Despite this large MUE of the originally evaluated energies by GFN2-xTB computations, a remarkable improvement in the accuracy of predicted energies was achieved via the developed machine-learning models based on different combinations of ImPerHam representations. According to the results, the studied machine-learning models could yield an MUE of 1.40 kcal/mol and RMSE of 2.47 kcal/mol via a neural network model taking 38 ImPerHam representations as model inputs and 16 neurons in the hidden layer. These results show an improvement in originally computed energies based on the GFN2-xTB method by one order of magnitude. A comparison of the reference CCSD(T) configuration energies and the machine-learned and GFN2-xTB computed ones are depicted in Fig. 2. As a further illustration, for the three dimers with the greatest energy variations between their conformers, the CCSD(T) energies are compared with machine learning and GFN2-xTB evaluated energies in different conformers in Fig. 3.

**Machine-learning estimation of solvation free energies via ImPerHam representations**. Among the studied machine-learning models for estimation of the solvation free energies, the best result was obtained for a neural network with only 5 neurons in the hidden layer and 22 ImPerHam representations and solvent dielectric constants as model inputs. For this model, we obtained an MUE of 0.545 kcal/mol. Compared to the accuracy of the original ALPB, for which an MUE of 1.4 kcal/mol has been reported[48], our results show a substantial improvement by almost a factor of three. Further comparison of the obtained results with conventionally accepted continuum solvation models is reported in Table 1. According to these results, the machine-learning evaluation of the solvation free energy via ImPerHam representations is significantly more accurate than SMD, PCM, and CPCM as the most extensively applied continuum solvation models, though obtained for a computational cost reduced by

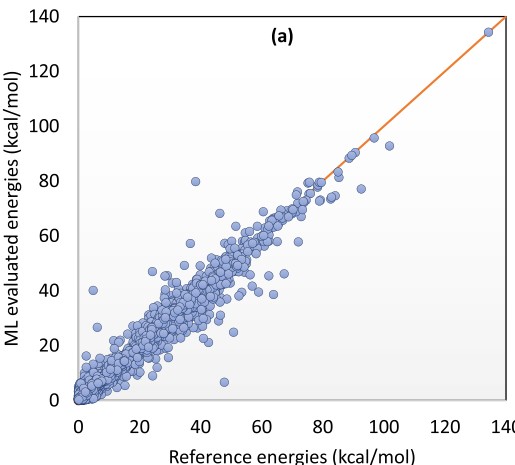
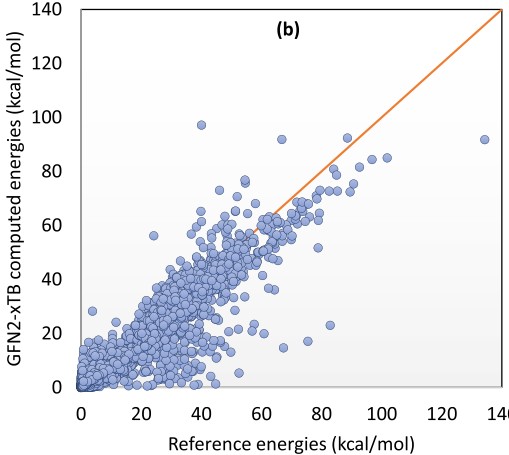

**Fig. 2 Comparison of reference CCSD(T) and evaluated energies.** A comparison of the reference energies with energies evaluated by machine learning based on ImPerHam representations (**a**) and computed by GFN2-xTB method (**b**) shows that the machine learning-based method results in a remarkably better agreement with the reference data.

several orders of magnitude (a few seconds compared to several minutes to hours on a normal desktop PC). The distribution of solvation free energies predicted via the newly proposed machine-learning method and by the SMD solvation model, in comparison to experimentally determined data, is depicted in Fig. 4.

Among the other solvation models reported in Table 1, only for the commercially available COSMO-RS solvation model higher accuracies have been reported. However, it should be noted that the results reported here are presented only as proof of concept. By trying more extensive sets of solvents, larger neural networks, and more demanding training of neural network models to search for the global minimum in the evaluated MUE, more accurate results by the machine-learning models are expected to be achievable.

**Analysis of the performance of studied solvents and representations.** As discussed earlier, we considered the gas medium as well as 9 solvents, which provided a diverse range of dielectric constants and therefore different perturbations of the molecular Hamiltonians. The resulting energy attributes were considered as inputs for machine-learning models. The studied representations were also a very limited subset of potential energy attributes that could be defined and used as molecular representations. By analysis of those developed models that were among the top 10% of all studied models in terms of accuracy, we investigated the performance of individual solvents and representations.

The percentage of presence of each one of the studied mediums in the selected models with the highest accuracies is depicted in Fig. 5.

As can be seen in Fig. 5, the most effective perturbing mediums are not the same in different case studies. For example, while the energy attributes perturbed by benzene are available in 96% and 92% of the most effective machine-learning models developed for evaluating molecular energies and CYP450 inhibition, respectively, they are present in only 30% of the models which are effective in the estimation of solvation free energy. Additionally, the unperturbed Hamiltonian energy attributes as well as those obtained via perturbed Hamiltonian by ethyl acetate, water, and octane, are the most efficient ones in all considered case studies. The other interesting observation here is that all the studied solvents are employed in at least one-third of the models.

The percentage of presence of considered energy attributes in the selected models is reported in Table 2. Similar to the studied

solvents, here also these results show significant diversity in the efficiency of various energy attributes depending on the application. For example, while the representation with id equal to 2 is present in 87.5% of the accurate models developed for evaluation of molecular energies, it has not been employed in any of the selected models for evaluation of solvation free energy or inhibition of CYP450. Similarly, in all case studies, we can find solvents or representations which are present in all of the selected models. Therefore, studying a wider range of potential solvents or representations gives both high flexibility of ImPerHam representations in studying various problems and allows for achieving higher accuracies.

As can be seen in Fig. 5, accurate prediction of solvation free energy by machine-learning models can be achieved for a significantly lower number of solvents. The potential reason for that can be attributed to the fact that solvation-free energy mainly depends on the dielectric constant of the solvent and the geometrical shape of the solute molecules[49]. While the former is already present in all machine-learning models as model input, as discussed in the method section, the latter remains the same for different solvents. For the same reason, all of the machine-learning models that yield the best results for evaluation of solvation free energy employ total Gibbs free energy and cavitation Gibbs free energy as model inputs, as can be seen in Table 2.

On the other hand, for the prediction of interaction energy, the potential energy attributes, such as isotropic electrostatic energy and the norm of the gradient vector, which is a measure of the energy difference between the molecular structure under study and the most stable conformation, are obviously the most relevant representations to the interaction energy. Consequently, these energy attributes are present in 100% of the machine-learning models that yield the highest accuracies for the evaluation of conformational energies. Additionally, unlike the geometrical shapes required by the evaluation of solvation-free energy, such energy attributes vary more significantly in different solvents. As a result, the highest employment of all studied solvents also is observed for this application.

For inhibition of the CYP450, as it is implied from the physics of the problem, the free energy of solvation of drug candidates in different solvents can determine the free energy of docking the drugs to the active site of the enzyme[50,51]. To that end, the most widely considered solvents are octanol and water, commonly

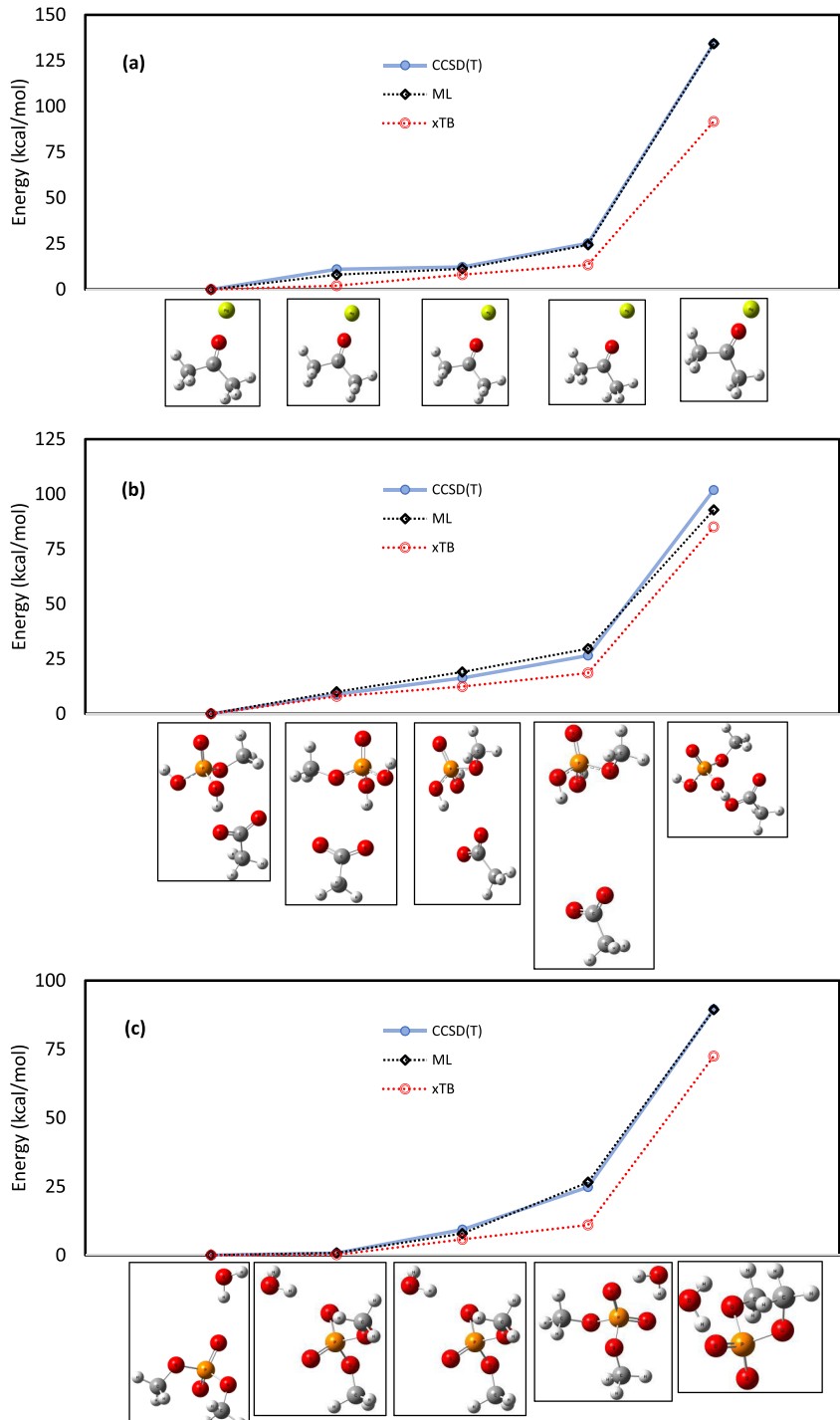

**Fig. 3 Comparison of reference CCSD(T) energies and predicted energies in different conformers.** For three dimers (**a**), (**b**), and (**c**) which showed the highest variability in energy range among conformers, employing machine learning and ImPerHam representations remarkably improves the agreement between predicted and reference data.

employed as octanol-water partition coefficient[52–56]. Very interestingly, these two solvents are also present in all of the developed machine-learning models with the highest efficiency in evaluating the CYP450 inhibition. For the same reason, the energy attributes which are most clearly related to the solvation free energy, such as electric, cavitation, and hydrogen bonding components of the solvation free energy, are present in 100% of the most effective machine-learning models developed for this specific application. From that perspective and considering that these energy attributes construct the total solvation free energy, the representation encoding the total solvation free energy becomes redundant and is not employed as extensively as its constituting components.

To summarize, in the present study, we introduced ImPerHam as highly versatile representations for describing molecular systems and developing advanced machine-learning models. The ImPerHam representations are various energy attributes that are computed via molecular Hamiltonians in vacuum as well

**Table 1 Comparison of the results of the new method with other models for solvation free energy prediction.**

| Method | Source | Nr. Samples | Nr. Solvents | Nr. Solutes | Deviation measure | Deviation (kcal/mol) |
|---|---|---|---|---|---|---|
| Machine learning | Present study | 2493 | 91 | 435 | MUE | 0.55 |
| | | | | | RMSE | 0.74 |
| Machine learning | Alibakhshi and Hartke[39] | 2224 | 88 | 300 | MUE | 0.24 |
| | | | | | RMSE | 0.37 |
| Machine learning | Vermeire and Green[38] | 10145 | 291 | 1368 | MUE | 0.21 |
| | | | | | RMSE | 0.44 |
| Machine learning | Weinreich et al.[37] | 642 | —— | —— | MUR | 0.57 |
| | | | | | R | 0.95 |
| COSMO-RS | Klamt and Diedenhofen[68] | 2346 | 91 | 318 | MUE | 0.42 |
| | | | | | RMSE | 0.70 |
| SM12 | Marenich et al.[69] | 2403 | 91 | 352 | MUE | 0.55-0.67 |
| DCOSMO-RS | Klamt and Diedenhofen[68] | 2346 | 91 | 318 | MUE | 0.66 |
| | | | | | RMSE | 1.00 |
| Feature Functional Theory | Wang et. al.[70] | 668 | 1 (water) | 668 | RMSE | 1.05 |
| kernel-based machine learning | Rauer and Bereau[71] | 355 | 1 (water) | 355 | MUE | 1.06 |
| atoms-in-molecules neural network | Zubatyuk et.al.[72] | —— | —— | 414 | MUE | 1.1 |
| Structure-Property Relationship | Hutchinson and Kobayashi[73] | —— | 1 (water) | —— | RMSE | 1.65 |
| SMD | Present study | 2493 | 91 | 435 | MUE | 0.79 |
| | | | | | RMSE | 1.16 |
| CPCM | Present study | 2493 | 91 | 435 | MUE | 2.69 |
| | | | | | RMSE | 3.17 |
| PCM | Present study | 2493 | 91 | 435 | MUE | 2.91 |
| | | | | | RMSE | 3.39 |

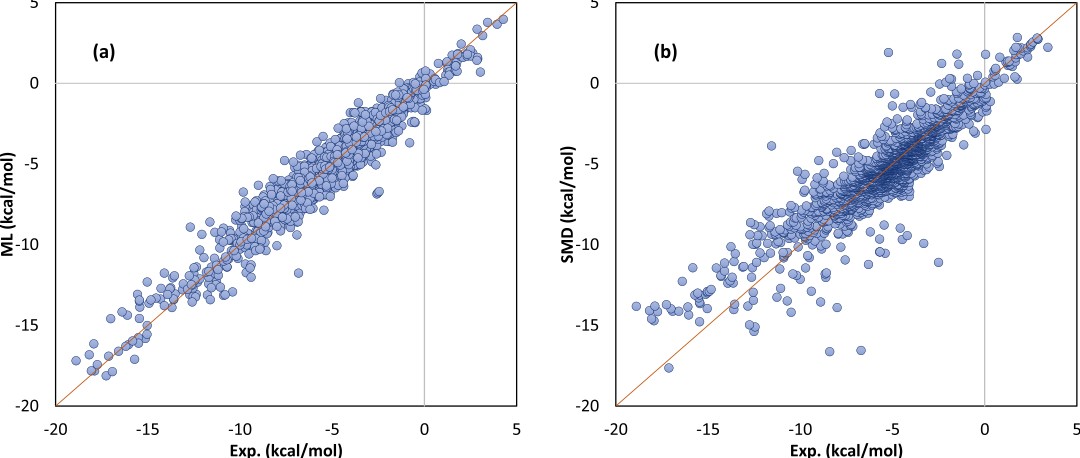

**Fig. 4 Comparison of predicted and reference solvation free energies.** The solvation free energies predicted via ML (**a**) are in better agreement with the reference data in comparison to the SMD method (**b**).

as in implicitly defined solvents. We demonstrated high efficiency and accuracy of machine-learning models based on ImPerHam representations in three diverse applications of predicting CYP450 inhibition by candidate molecules, evaluating the conformational energies in multi-atomic systems, and solvation free energies of diverse solute-solvent mixtures. Our results have shown the capability of ImPerHam representations in developing transferable machine-learning models applicable to diverse molecular systems.

## Methods
**Benchmark sets.** To be able to make a reliable evaluation of the newly proposed representations, we employed large benchmark sets commonly applied for each studied application.

Accordingly, for studying the inhibition of the CYP450 enzyme we exploited the benchmark set provided by Novotarskyi and co-workers for comparing different machine-learning models proposed for the same purpose[46]. We used the same compounds considered by them as training and test datasets to train and validate our model, which included 3745 and 3740 samples, respectively. Note that this benchmark data re-use implies that we have not done any molecule-protein docking calculations ourselves, but instead employed machine learning based on our ImPerHam representations to reproduce the inhibitor/non-inhibitor classification contained in this benchmark set.

To benchmark our newly proposed representations in reproducing high-quality molecular energies, we exploited CCSD(T) reference energies computed for multiple configurations of 3525 dimers provided in the DES370K database[57]. To reduce the computational cost of model development, we randomly selected roughly half of the available dimers a priori, which yielded a dataset containing 1155 neutral, 390 positively charged, and 177 negatively charged dimers. From the available conformers of each one of the selected dimers, five were selected for developing the machine-learning models, linearly distributed between the highest and lowest energy conformers.

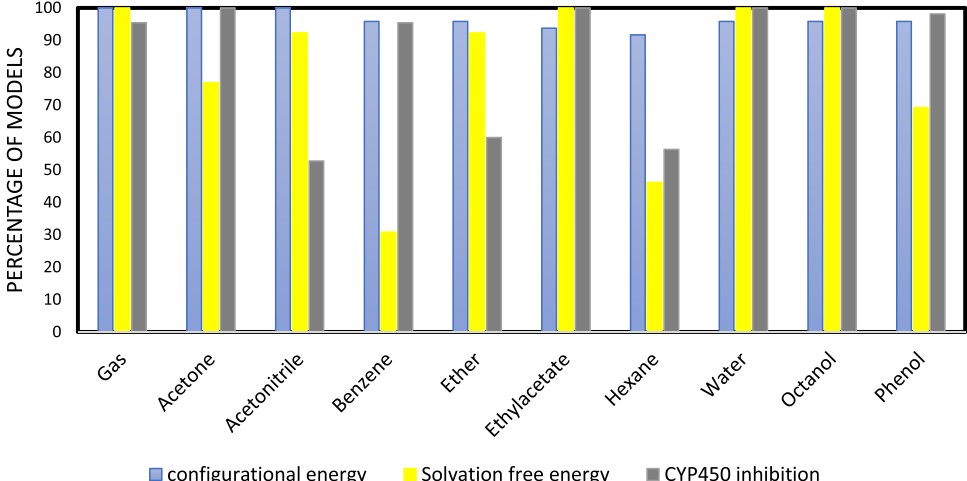

**Fig. 5 Percentage of the presence of studied mediums in selected models for different considered applications.** Perturbation of Hamiltonian by different solvents might have different impacts on predictability depending on the property of interest.

**Table 2 Percentage of the presence of studied energy attributes in selected models in different applications.**

| Representation ID | Conformational energy | Solvation free energy | CYP450 inhibition |
|---|---|---|---|
| Total energy | 87.50 | 15.38 | 25.45 |
| Total free energy (without cavity and hydrogen bonding contributions) | 87.50 | 0.00 | 0.00 |
| HOMO-LUMO gap | 16.67 | 61.54 | 71.82 |
| HOMO orbital eigenvalue | 79.17 | 84.62 | 71.82 |
| LUMO orbital eigenvalue | 79.17 | 38.46 | 100.00 |
| SCC energy | 75.00 | 46.15 | 0.00 |
| Gibbs free energy (total) | 93.75 | 100.00 | 57.27 |
| Gibbs free energy (electric) | 37.50 | 15.38 | 100.00 |
| Gibbs free energy (cavitation) | 100.00 | 100.00 | 100.00 |
| Gibbs free energy (hydrogen bond) | 45.83 | 92.31 | 100.00 |
| Free energy shift for infinite dilution | 16.67 | 0.00 | 0.00 |
| Gradient norm | 100.00 | 84.62 | 44.55 |
| Isotropic electrostatic energy | 100.00 | 15.38 | 46.36 |
| Anisotropic exchange correlation | 79.17 | 69.23 | 71.82 |
| Isotropic exchange correlation | 16.67 | 15.38 | 97.27 |
| Dispersion energy | 37.50 | 92.31 | 44.55 |
| Repulsion energy | 8.33 | 46.15 | 13.64 |
| Atomization energy | 79.17 | 100.00 | 0.00 |

For studying the predictability of solvation free energies, the efficiency of the developed machine-learning models was benchmarked using the Minnesota solvation database[58] containing 2493 reference solvation free energy data for binary mixtures of 435 solutes and 91 solvents.

Details of the studied samples and computed representations for each one are provided as supplementary materials.

The accuracy of the developed models is reported as the mean unsigned error (MUE) and root mean square error (RMSE) in kcal/mol, defined as

$$\text{MUE} = \frac{1}{N} \Sigma \left( \left| y_i^{\text{exp}} - y_i^{\text{pred}} \right| \right), \quad (1)$$

$$\text{RMSE} = \sqrt{\left( \frac{1}{N} \Sigma \left( y_i^{\text{exp}} - y_i^{\text{pred}} \right)^2 \right)}. \quad (2)$$

**Generating representations**. We generated ImPerHam representations by perturbing the Hamiltonian via the analytical linearized Poisson-Boltzmann (ALPB) solvation model[48], as implemented in Grimme's Semiempirical Extended Tight-Binding Program Package[59] (xTB). The quantum-mechanical computations were carried out with the GFN2-xTB semiempirical method[60], for fast and computationally inexpensive acquisition and reasonable estimation of the studied energy attributes.

For each molecule in the vacuum state as well as in implicitly defined acetone, acetonitrile, benzene, hexane, water, ether, ethyl acetate, 1-octanol, and phenol solvents, the Hamiltonian energy attributes listed in Table 3, were computed with original parameterizations as provided in xTB[59]. This constituted the initial pool of representations.

These solvents were selected from a wider choice of available solvents parameterized in xTB, with the intention to span a diverse range of dielectric constants. Nevertheless, any real or hypothetic solvent with an arbitrary or actual dielectric constant can also be used for this purpose, resulting in a wide range of relevant representations and higher flexibility of the machine-learning models to evaluate different quantities. This limited number of solvents was considered only to present the approach and efficiency of the proposed representations and as proof of concept.

For the computed Hamiltonians, the energy attributes reported in Table 3 were considered as potentially relevant representations. Nevertheless, many other relevant energy attributes can also be defined and employed for this purpose.

For the machine-learning models developed to approximate conformational energies via ImPerHam representations, among the 5 selected configurations of each individual dimer, the conformation with the lowest energy was considered as the reference and its energy was set to zero. The energies of the other conformations of the same dimer were also corrected accordingly. This merely is the standard practice in most computational chemistry applications, to get away from total energies (that are irrelevant in most cases, and also frequently method-dependent). Here it focuses the machine-learning approach on the relevant relative energies between conformers.

By the above-mentioned treatment of the CCSD(T) conformational energies, the computed representations also were corrected the same way, i.e. the deviations of computed representations for each conformer with respect to the one with the lowest total energy were considered as the inputs to the machine-learning models.

For evaluation of solvation free energies, considering that the reference data were defined as the difference between the free energies of two liquid and gas states, we did not employ the same modification of energy and representations. However, in addition to the computed ImPerHam representations, we also considered the

**Table 3 List of energy components of the implicitly perturbed Hamiltonian employed as molecular representations.**

| | |
|---|---|
| 1 | Total energy |
| 2 | Total free energy without cavity and hydrogen bonding contributions |
| 3 | HOMO-LUMO gap |
| 4 | HOMO orbital eigenvalue |
| 5 | LUMO orbital eigenvalue |
| 6 | SCC energy |
| 7 | Gibbs free energy (total) |
| 8 | Gibbs free energy (electric) |
| 9 | Gibbs free energy (cavitation) |
| 10 | Gibbs free energy (hydrogen bond) |
| 11 | Free energy shift for infinite dilution |
| 12 | Gradient norm |
| 13 | Isotropic electrostatic energy |
| 14 | Anisotropic exchange correlation |
| 15 | Isotropic exchange correlation |
| 16 | Dispersion energy |
| 17 | Repulsion energy |
| 18 | Atomization energy |

dielectric constant of solvents in each solute-solvent mixture as additional model input, to characterize the specific solvent under study, similar to conventional continuum solvation models and our recently developed machine-learning model[39].

To compare the accuracy of solvation free energies obtained by machine learning with those obtained via conventional solvation models, we also computed the solvation free energies via the widely accepted SMD[61], PCM[62], and CPCM[63] continuum solvation models at the B3LYP/6-311 + g(2d,p) level of theory in Gaussian 16[64]. To that end, the geometry of the solutes was first optimized and then used for the computation of solvation free energy.

**Developing machine-learning models**. For more efficient and tractable training of machine-learning models, we employed variable selection as a commonly used approach in developing machine-learning models. To that end, the initial pool of computed representations was screened employing the MRMR variable selection algorithm[65] to yield different sets of input variables containing 4 to 45 representations.

After computation and screening of the required representations based on the above recipe, the machine-learning models taking those representations were set up to study the applications of interest.

Considering that the inhibition of CYP450 enzyme by potential drug candidates is a classification problem, i.e. the role of machine learning is only to classify the studied molecules as inhibitor or non-inhibitor, we employed the Support-Vector-Machine (SVM) classification[66] as a rigorous machine-learning method developed for this purpose. To that end and to be able to make a reliable comparison to the large number of machine-learning models studied by Novotarskyi et al.[46] for the same purpose, we used the same training and test samples employed in that work. Developing the SVM models was carried out by the fitcsvm module of Matlab[67] with the default settings.

For evaluation of interaction energy and solvation free energy, we employed artificial neural networks as a highly efficient machine-learning tool to map the dependency between required quantities and the proposed representations. To that end, training of the ANN models was carried out based on the guidelines provided in our recent study[47].

Accordingly, we assigned 60% of the dataset for training, 15% for validation and 25% for testing the machine-learning models. We only studied small neural networks with one hidden layer and a very limited number of neurons. The upper limit of the selected number of neurons was assigned to allow having roughly 10 training samples or more per ANN constant. In many neural network models, much lower ratios of training samples to ANN constants are commonly employed, which can result in much higher accuracies for the obtained results. However, this may lead to reducing the extrapolation capability of the neural networks[47]. Therefore, we only considered a limited number of neurons with the upper limit discussed above. We initially tried 40 different dataset divisions and for each one 100 randomly assigned initial values for the ANN weight and bias constants. For each model which in initial training yielded MUE lower than the best found results, to verify that the good result is not due to overfitting, we took the optimized ANN parameters as initial guesses and retrained those models under 100 different and random divisions of the dataset into training, validation and test sets. For each one of these replicas, we compared the MUE of the training and validation set by standard t-test method and selected a model as reliably trained one if in at least 80% of all the replicasit resulted in no significant difference between the MUE of the training and test sets at 95% of significance level 59.

For the machine-learning models developed to evaluate conformational energies, training, validation, and test sets were assigned based on the individual dimer types and not individual conformers, to avoid unreliable high-accuracy results due to interpolation. In other words, if a dimer was assigned to a training, validation, or test set, all its conformers were also assigned to the same set.

## Data availability

All data produced in this study are available and can be provided by contacting the corresponding author.

## Code availability

The source file of the C++ code developed for implementing the proposed method with detailed used instructions are available as supplementary material or can be provided by contacting the corresponding author.

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

## Author contributions

A.A. has conceived and developed the presented methods, carried out all the computations and has had the major contribution in writing the manuscript. B.H. has contributed in interpreting the results and physicalsignificance of imperham representations, discussions, and writing the manuscript.

## Funding

## Competing interests

The authors declare no competing interests.
