## [Peer Review File · Nature Communications]

REVIEWER COMMENTS

Reviewer #1 (Remarks to the Author):

This work provides a new molecular representation for machine learning purposes. The molecular representation is based on features determined by low-level quantum calculations. The new representation is used on some common datasets including for the prediction of enzyme inhibition, dimer conformational energy, and solvation free energy. Overall the work is sound and can be valuable to the community. Although there are some aspects that should be further explained. Detailed comments are given below.

1. Can the code be made available to the community? Such that the ImPerHam representation can be reproduced and used in other work?
2. P3. The authors say that machine learning for solvation free energy have results that are still beyond chemical accuracy. However, two recent studies actually showed that machine learning models can perform better than the experimental uncertainty present in solvation free energy datasets. (<https://doi.org/10.1016/j.cej.2021.129307>, <https://doi.org/10.1063/5.0041548>)
3. The computational cost of generating ImPerHam representations is mentioned as one of the downsides of this work. Can this be quantified? How long does it take for e.g. a dataset of 1000 molecules?
4. The MNSol database is used as a benchmark for solvation free energy. However, recently larger and open source datasets were published for solvation free energy that cover more solutes and solvents. ([10.33774/chemrxiv-2021-djd3d](https://doi.org/10.33774/chemrxiv-2021-djd3d))
5. The model performance is assessed with the MUE. However, additional information on the RMSE (root-mean-squared-error) can provide more and valuable information on outliers.
6. For generating the ImPerHam representation it is not completely clear how different conformations of the molecules are accounted for. For the prediction of the conformational energy, I assume only one conformer is accounted for. For enzyme inhibition one conformational structure is important, while for solvation free energies a set of conformations is important. Hence, the

conformations that should be accounted for depend on the property that has to be predicted. Can the authors clarify which conformers are accounted for and how they are generated?

7. The reviewer does not agree that the use of a dielectric constant is sufficient to represent a solvent as claimed by the authors (as also stated on p15). This is clearly shown by a different performance of SMx and COSMO-RS as quantum chemical methods. Also using machine learning, an explicit solvent representation has significant benefits compared to the single use of a dielectric constant.

8. It is not clear what the set of 45 features are used for the feature selection. Tale 1 only shows 18 features.

9. More information is needed on the SVM and ANN. How are these models optimized. What are the hyperparameters and how are they selected. Readers should be able to reproduce the results based on the information provided in the manuscript.

10. Can you compare the results for CYP450 inhibition to the QSAR published by Novotarskyi et al. by means of MUE and RMSE?

11. The performance for the solvation free energy is worse compared to many recently published machine learning models for solvation free energies. In many cases, the different performance of the model can be attributed to different training and test sets. However it would be good to compare the performance of the new representation to the most recent published ML models.

12. For Table 3 it would be better to describe the representation instead of giving the ID.

Reviewer #2 (Remarks to the Author):

The authors present in this contribution a promising set of features for the description of molecular systems under machine-learning (ML) approaches. In the very first applications, and without

extensive comparison of different ML models, the results already seem quite promising. The added computational cost compared to other feature sets is not dramatic, but at some points the comparison should be made in a more balanced way (see below). I believe this is a valuable addition to the descriptors' toolbox of computational chemistry and will be of interest to the community. The range of applications is also quite adequate to show the potential of the features. However, I do still have some comments on the text.

- in page 2, when noting the limitation of chemical accuracy of ML models to smaller molecular systems, one should also note the developments in molecular dynamics and spectroscopic quality of ML-based potential surfaces. A good reference to add would be the very recent 2021 Chem. Rev. 10.1021/acs.chemrev.0c00665 from Carrington.

- It is not clear how the features are standardized. The authors deflect all details to a previous publication, but some essential information should be included, and under this I believe the pre-processing of the features is quite relevant.

- I do not agree with the use of MUE as single quality indicator. Given how important these accuracy considerations are for assessing the relevance of the proposed features, the authors should take into account the discussion of Pernot and Savin on this topic (10.1007/s00214-021-02725-0). An indicator (authors' choice) quantifying the risk of larger errors would be very useful.

- regarding the evaluation of CYP450 inhibition results, I find the discussion should be somewhat improved. First of all, the best models should be explicitly mentioned and compared. On top, some more recent papers on the classification of CYP450 inhibitors should be cited (or effectively argued why these should not be included). The publications 10.1021/acs.molpharmaceut.8b00110 and 10.1021/acs.jcim.5b00130 show accuracies close to and beyond 90%. Underscoring such results on the basis of the number of representations is not totally fair, since one could potentially make use of variable selection procedures to reduce said number. The authors do not have enough information to judge if this an actual issue.

minor points

- Fig. 4 is referenced before Fig. 3. The ordering should be changed.

- caption of Fig. 2 and text. I do not see any reason to show 3 decimal places in the MUE.

Reviewer #3 (Remarks to the Author):

Summary

The paper proposes a new molecular fingerprint to be used in machine learning (ML) regression and classification. The fingerprint is mostly based on energetic quantities computed with simple and cheap models, which are more or less related to the predicted quantities. The authors use this fingerprint as features for SVM and NN models to predict binding and interaction/solvation energies and reach accuracies comparable or better than the state of the art.

General comments

The idea is interesting. Although much more expensive to compute than other types of fingerprints (for example based on structural information) it is also more informative, and it might prove useful in several applications. The paper is well written and the authors provide code and data to reproduce the results.

I only have a few minor suggestions, but I believe the paper will be a nice addition to the literature and I recommend publication after my concerns have been addressed.

Minor suggestions

- Please do include confidence intervals for the metric statistics. A simple 95% confidence interval obtained by bootstrapping will do but simply providing the MUE of the ML-method and the baseline without an estimate of the uncertainty makes it hard to understand how much the comparison is statistically robust.

- I seem to remember that the DES370K dataset provides noncovalent interaction energies. If this is the case, please consider specifying this instead of using "conformational energies", which often refers to the potential energy that takes into account intramolecular contributions.

- There is a duplicate reference: 36 and 37.

Dear Reviewers,

On behalf of myself and my co-author, I would like to thank you very much for your time for considering our work. We also appreciate the careful review and fruitful comments of the reviewers and found them very helpful in improving the quality of our work.

In the following, we have provided point-by-point responses to the reviewers' comments.

We hope to have addressed all major concerns with our responses and revisions. We would be happy and prepared to provide further clarifications and appropriate revision if required.

Best regards,

Amin Alibakhshi

=====

REVIEWER COMMENTS

Reviewer #1 (Remarks to the Author):

This work provides a new molecular representation for machine learning purposes. The molecular representation is based on features determined by low-level quantum calculations. The new representation is used on some common datasets including for the prediction of enzyme inhibition, dimer conformational energy, and solvation free energy. Overall the work is sound and can be valuable to the community. Although there are some aspects that should be further explained. Detailed comments are given below.

=====

Comment: 1. Can the code be made available to the community? Such that the ImPerHam representation can be reproduced and used in other work?

Response: For the current work, we provided a code for user-friendly implementation of the developed machine learning models, which currently still requires generating the representations via QM computations using the xTB code. Nevertheless, it is our definite plan in a near future, for each one of the specific applications, to provide an open-source code with GUI for automating all the computations.

=====

Comment: 2. P3. The authors say that machine learning for solvation free energy have results that are still beyond chemical accuracy. However, two recent studies actually showed that machine learning models can perform better than the experimental uncertainty present in solvation free energy datasets. (<https://doi.org/10.1016/j.cej.2021.129307>, <https://doi.org/10.1063/5.0041548>)

Response: We also totally agree that some of the recent ML models, specifically the two interesting works mentioned in your comment, provide quite good results for the prediction of the solvation free energy. Nevertheless, except for these very recent works, still the majority of the ML models provide results that are beyond chemical accuracy. Therefore, in our initial version of the manuscript we declared:

“the results obtainable via *most* of the currently developed models are still beyond chemical accuracy”.

Among the two recent ML works, we already mentioned the work of Vermeire and Green in table 2 in our initial submission. The other ML model of von Lilienfeld’s group was however missing in our literature survey which is now considered in our comparison provided in table 2 as well as a specific comment on the two mentioned works in our revised manuscript, based on your comment (cf. page 7, paragraph 4)

=====
Comment: 3. The computational cost of generating ImPerHam representations is mentioned as one of the downsides of this work. Can this be quantified? How long does it take for e.g. a dataset of 1000 molecules?

Response: As a matter of fact, what we declared as higher computational cost of our computations was in comparison to geometry-based representations. Nevertheless, compared to even medium level full QM computations, e.g. for implementing classical continuum solvation models at the DFT level of theory, the computational cost of the provided ML models is lower by several orders of magnitude. We cannot provide a reliable estimation for the computational time required for a dataset of 1000 molecules because it can depend on the size of each molecule and the computational power. However, to provide a rough impression about the computational efficiency, we can mention our experience which for our studied systems on a single node with 32 cores and 40 GB of memory, for small to medium sized molecules, the computations (low-cost QM followed by ML computations) didn’t take more than very few seconds for each molecule (in most cases in fact only a fraction of a second).

=====
Comment: 4. The MNSol database is used as a benchmark for solvation free energy. However, recently larger and open source datasets were published for solvation free energy that cover more solutes and solvents. (10.33774/chemrxiv-2021-djd3d)

Response: Considering that the main aim of our work was to introduce the new representations and show their efficiency and flexibility by early results in some applications and only as a proof of concept as we declared in the manuscript, we didn’t focus so much on obtaining the best possible results for each application (which will be the focus of our future studies) and consequently, a critical comparison of our model with other works. Accordingly, we employed the MNSol database as a standard database which has been extensively used in developing solvation models.

=====
Comment: 5. The model performance is assessed with the MUE. However, additional information on the RMSE (root-mean-squared-error) can provide more and valuable information on outliers.

Response: We totally agree and it was also the reason that in our initially submitted manuscript, we also have reported RMSE for results obtained for prediction of the solvation free energy (cf. table 2). In our revision, we clarified it in the method section and also have provided the RMSE for the other results.

=====
Comment: 6. For generating the ImPerHam representation it is not completely clear how different conformations of the molecules are accounted for. For the prediction of the conformational energy, I assume only one conformer is accounted for. For enzyme inhibition one conformational structure is important, while for solvation free energies a set of conformations is important. Hence, the conformations

that should be accounted for depend on the property that has to be predicted. Can the authors clarify which conformers are accounted for and how they are generated?

Response: This is indeed a very good point. For prediction of the interaction energies or CYP450 inhibition, as you also noticed, we were restricted to a single conformation which was used also to obtain the reference data. For the solvation free energy, we totally agree that in reality, multiple conformations contribute to the entropy and free energy and for properly taking into account the physics behind the problem, multiple conformers should be considered, not only for different conformations of the solute but also for different configurations of solute and explicitly defined solvent molecules as a cluster, as we elaborated in a recent study (DOI: 10.33774/chemrxiv-2021-lpdnx). Nevertheless, for computations based on implicit solvent models, studying multiple conformations is not typically required as they are implicitly taken into account by parameterization of the solvation models as we also discussed and demonstrated in the recent work mentioned above.

=====
Comment: 7. The reviewer does not agree that the use of a dielectric constant is sufficient to represent a solvent as claimed by the authors (as also stated on p15). This is clearly shown by a different performance of SMx and COSMO-RS as quantum chemical methods. Also using machine learning, an explicit solvent representation has significant benefits compared to the single use of a dielectric constant.

Response: Totally agree. Nevertheless, as we discussed in response to the previous comment, in continuum solvation models, description of solvent by a single parameter (the dielectric constant) and taking into account the inaccuracies due to insufficient description of the solvent this way by adjustable parameters (either in classic solvation models as conventionally done or by adjusting weight and bias constants in ML models) still can be considered as a method which can yield reliable estimation of the solvation free energy.

=====
Comment: 8. It is not clear what the set of 45 features are used for the feature selection. Tale 1 only shows 18 features.

Response: The 18 features are 18 energy attributes of the Hamiltonian. Considering that they are computed for 9 different studies solvents and also for in vacuo, it results in a total number of 180 representations for each molecule. Among those 180 computed quantities we have selected 45 of them for the present model.

=====
Comment: 9. More information is needed on the SVM and ANN. How are these models optimized. What are the hyperparameters and how are they selected. Readers should be able to reproduce the results based on the information provided in the manuscript.

Response: Considering that so much details on individual ML models and intricacies of their development can make the work too lengthy and is not also needed for implementing the developed models and reproducing the results considering that we have already provided the codes for their application (it includes all the parameters and hyperparameters) and also considering that such training intricacies are not unique and with other arbitrary optimization techniques, a similar or even better results might be obtained, we refrained to provide specific details in the text and referred the readers which are interested to develop new ML models to the given references. Nevertheless, some important details on training the ML models are provided in our revision in section 2-3.

=====

Comment: 10. Can you compare the results for CYP450 inhibition to the QSAR published by Novotarskyi et al. by means of MUE and RMSE?

Response: We provided more detailed comparisons in section 3-1.

=====
Comment: 11. The performance for the solvation free energy is worse compared to many recently published machine learning models for solvation free energies. In many cases, the different performance of the model can be attributed to different training and test sets. However it would be good to compare the performance of the new representation to the most recent published ML models.

Response: That's totally true. And in addition to the differences resulted by different benchmark sets, the size of the ML model also plays a very important role in the obtained accuracy. As we demonstrated in figure 1 in our previous study (Nature Communications, volume 12, Article number: 3584 (2021)), by increasing the number of hidden layer neurons, the MUE usually drops rapidly. The results we report are obtained for a very limited size of the system. For example, for prediction of the solvation free energy our selected model includes only one hidden layer with only 5 neurons which results in only 121 parameters for the ANN model which is a much smaller number of parameters than used in most of the other ML models (e.g. ca 700K parameters required by the ML models of Vermeire and Green [DOI:10.1016/j.cej.2021.129307](https://doi.org/10.1016/j.cej.2021.129307) developed for the same purpose). Additionally, as we also mentioned in response to your 4th comment above, our main aim has not been to provide the best obtainable results by our newly proposed representations. As we also discussed in the manuscript, we only studied 10 solvents and a limited number of energy attributes and provided those results only as a proof of concept. By studying a larger set of solvents or energy attributes as well as with more focus on optimization details, obtaining better results might be achievable which will be the subject of future studies. Nevertheless, we already provided some comparisons with different models and have extended the studied works in our comparison in table 2.

=====
Comment: 12. For Table 3 it would be better to describe the representation instead of giving the ID.

Response: We made this change in our revised manuscript according to your comment.

=====
=====
Reviewer #2 (Remarks to the Author):

The authors present in this contribution a promising set of features for the description of molecular systems under machine-learning (ML) approaches. In the very first applications, and without extensive comparison of different ML models, the results already seem quite promising. The added computational cost compared to other feature sets is not dramatic, but at some points the comparison should be made in a more balanced way (see below). I believe this is an valuable addition to the descriptors' toolbox of computational chemistry and will be of interest to the community. The range of applications is also quite adequate to show the potential of the features. However, I do still have some comments on the text.

=====
Comment: 1- in page 2, when noting the limitation of chemical accuracy of ML models to smaller molecular systems, one should also note the developments in molecular dynamics and spectroscopic

quality of ML-based potential surfaces. A good reference to add would be the very recent 2021 Chem. Rev. 10.1021/acs.chemrev.0c00665 from Carrington.

Response: Thanks for providing this interesting reference. We added this work in our literature survey in the revised version of the manuscript.

=====
Comment: 2- It is not clear how the features are standardized. The authors deflect all details to a previous publication, but some essential information should be included, and under this I believe the pre-processing of the features is quite relevant.

Response: Please see the response to comment Nr. 9 of reviewer #1. Nevertheless, we provided further details on developing the models in section 2-3.

About pre-processing, it was true and we standardized the representations with the MapMinMax algorithm in Matlab.

=====
Comment: 3- I do not agree with the use of MUE as single quality indicator. Given how important these accuracy considerations are for assessing the relevance of the proposed features, the authors should take into account the discussion of Pernot and Savin on this topic (10.1007/s00214-021-02725-0). An indicator (authors' choice) quantifying the risk of larger errors would be very useful.

Response: We totally agree. According to this comment as well as the comment Nr. 5 of reviewer #1, we also reported RMSE for the obtained results.

=====
Comment: 4- regarding the evaluation of CYP450 inhibition results, I find the discussion should be somewhat improved. First of all, the best models should be explicitly mentioned and compared. On top, some more recent papers on the classification of CYP450 inhibitors should be cited (or effectively argued why these should not be included). The publications 10.1021/acs.molpharmaceut.8b00110 and 10.1021/acs.jcim.5b00130 show accuracies close to and beyond 90%. Underscoring such results on the basis of the number of representations is not totally fair, since one could potentially make use of variable selection procedures to reduce said number. The authors do not have enough information to judge if this an actual issue.

Response: As we also explained in response to comment Nr. 11 of reviewer #1, a careful and fair comparison of the efficiency of different representations and ML models can be made only if they are compared for the same benchmark set and size of the model. For the two mentioned works, we were not confident such a comparison would be reliable, specially because of the extremely large size of the ML models developed in those works. For example, in the publication 10.1021/acs.molpharmaceut.8b00110 as can be inferred from figure 3, it seems that this model includes roughly 6 million parameters which are optimized using only 13000 reference data, implying an underdetermined optimization problem. At such conditions, the risk of overfitting is quite high and for this reason, we refrained to include those works in our comparison.

The same can also happen for having a very large number of representations which is the reason we suggest requiring a much lower number of representations and ML parameters as advantages of the new set of representations.

=====
*Comment: 5- Fig. 4 is referenced before Fig. 3. The ordering should be changed.
- caption of Fig. 2 and text. I do not see any reason to show 3 decimal places in the MUE.*

Response: Thanks for noticing that. The incorrect ordering was due to automatic labeling in Latex which we corrected in our revision.

For the decimal places also, we reduced it for all the reported numbers to two decimal places.

=====
=====
Reviewer #3 (Remarks to the Author):

Summary

The paper proposes a new molecular fingerprint to be used in machine learning (ML) regression and classification. The fingerprint is mostly based on energetic quantities computed with simple and cheap models, which are more or less related to the predicted quantities. The authors use this fingerprint as features for SVM and NN models to predict binding and interaction/solvation energies and reach accuracies comparable or better than the state of the art.

General comments

The idea is interesting. Although much more expensive to compute than other types of fingerprints (for example based on structural information) it is also more informative, and it might prove useful in several applications. The paper is well written and the authors provide code and data to reproduce the results.

I only have a few minor suggestions, but I believe the paper will be a nice addition to the literature and I recommend publication after my concerns have been addressed.

Minor suggestions

=====
Comment: 1- Please do include confidence intervals for the metric statistics. A simple 95% confidence interval obtained by bootstrapping will do but simply providing the MUE of the ML-method and the baseline without an estimate of the uncertainty makes it hard to understand how much the comparison is statistically robust.

Response: It is indeed a good point. We actually employed t-test method at 95% to verify the robustness of our results statistically but we didn't clarify it in the text as we had referred it to our previous works. We added some more details and discussions on that in section 2-3.

=====
Comment: 2- I seem to remember that the DES370K dataset provides noncovalent interaction energies. If this is the case, please consider specifying this instead of using "conformational energies", which often refers to the potential energy that takes into account intramolecular contributions.

This is true and a very good point. We made the required changes in the revised version of the manuscript accordingly.

Response: That's true. We clarified it in our revision.

=====
Comment: 3- There is a duplicate reference: 36 and 37.

Response: Thanks for noticing. We corrected it in our revision.

REVIEWERS' COMMENTS

Reviewer #1 (Remarks to the Author):

In their comments, the authors addressed my concerns in detail and made some changes to the manuscript. After these changes and their explanation, I accept this manuscript for publication.

Reviewer #2 (Remarks to the Author):

The authors have properly addressed my comments. I would just like to note that in the resubmitted manuscript (in pdf form) Eq. (2) is not displaying properly.

Reviewer #2 (Remarks to the Author):

Comment: The authors have properly addressed my comments. I would just like to note that in the resubmitted manuscript (in pdf form) Eq. (2) is not displaying properly.

Response: Thanks for noticing. We fixed the displaying issue in our final check-up.